# Does Where You Live Predict What You Say? Associations between Neighborhood Factors, Child Sleep, and Language Development

**DOI:** 10.3390/brainsci12020223

**Published:** 2022-02-06

**Authors:** Queenie K. W. Li, Anna L. MacKinnon, Suzanne Tough, Susan Graham, Lianne Tomfohr-Madsen

**Affiliations:** 1Department of Psychology, University of Calgary, Calgary, AB T2N 1N4, Canada; queenie.li@ucalgary.ca (Q.K.W.L.); anna.mackinnon@ucalgary.ca (A.L.M.); grahams@ucalgary.ca (S.G.); 2Owerko Centre, Alberta Children’s Hospital Research Institute (ACHRI), Calgary, AB T3B 6A8, Canada; stough@ucalgary.ca; 3Department of Pediatrics and Community Health Sciences, Cumming School of Medicine, University of Calgary, Calgary, AB T2N 1N4, Canada

**Keywords:** child sleep, language development, neighborhood deprivation, neighborhood disorder

## Abstract

Language ability is strongly related to important child developmental outcomes. Family-level socioeconomic status influences child language ability; it is unclear if, and through which mechanisms, neighborhood-level factors impact child language. The current study investigated the association between neighborhood factors (deprivation and disorder) assessed before birth and child language outcomes at age 5, with sleep duration as a potential underlying pathway. Secondary analysis was conducted on data collected between 2008 and 2018 on a subsample of 2444 participants from the All Our Families cohort study (Calgary, Canada) for whom neighborhood information from pregnancy could be geocoded. Neighborhood deprivation was determined using the Vancouver Area Neighborhood Deprivation Index (VANDIX), and disorder was assessed using crime reports. Mothers reported on their children’s sleep duration and language ability. Multilevel modeling indicated that greater neighborhood deprivation and disorder during pregnancy were predictive of lower scores on the Child Communication Checklist–2 (CCC–2) at 5 years. Path analyses revealed an indirect effect of neighborhood disorder on language through child sleep duration at 12 months. These results add to growing evidence that child development should be considered within the context of multiple systems. Sleep duration as an underlying link between environmental factors and child language ability warrants further study as a potential target for intervention.

## 1. Introduction

The emergence of language during early childhood is a remarkable developmental accomplishment. Strong language skills are positively associated with self-regulation, social competence, and academic performance in children [1,2,3,4,5]. For example, language skills at the onset of formal education (i.e., around five years of age) strongly predict achievement and psychiatric health into late adolescence [6,7]. Conversely, poor language ability is linked with developmental consequences including difficulties with emotion regulation, internalizing and externalizing behavioral problems, as well as attention deficit/hyperactivity disorder [8,9,10]. The present study explored how neighborhoods influence child language development, and whether sleep represents a potential underlying mechanism.

Language delays have been observed in 6 to 20% of children in the first three years of life, with about one-third of those children not catching up to their peers [11,12]. As conceptions of healthy development have shifted focus to social and interpersonal concerns, language therapists argue that language disparities are a matter of public health that warrant population-level solutions [13]. Further, there exists a tension in contemporary efforts to understand and intervene in child language problems; deficit models of language learning broadly emphasize family-level socioeconomic differences, while strengths-based approaches encourage attention to structural inequity and a reliance on more culturally sensitive frameworks [14,15,16]. Thus, it is essential to study language skills in the context of multiple, layered environmental influences in order to advance equitable interventions with modifiable targets at both individual and population levels.

Bioecological models offer a useful framework to understand influences on language acquisition as they posit that child development occurs within nested layers of systems [17]. These developmental contexts range from proximal environments that have the largest impact (e.g., families) to distal environments that are less direct in their influence (e.g., neighborhoods). One factor that is commonly considered at both the family and neighborhood levels is socioeconomic status (SES). At the family level, SES refers to overall social standing as constrained by access to resources [18,19]. The relationship between low SES and child language deficits has preoccupied researchers for decades [20]. Children from lower-SES homes demonstrate difficulties in all areas of language including comprehension, production, and narrative function [21]. Disparities are observed in children as young as 18 months and, by two years, disadvantaged children have been shown to be up to six months behind in language acquisition compared with their more affluent peers [22]. Unsurprisingly, developmental problems related to language such as maladaptive social functioning and poor mental health are also more prevalent among children from lower SES families [23].

The robust link between family-level SES and development has led to the increasing relevance of neighborhood-level deprivation on child health and behavioral outcomes [24]. Neighborhood deprivation is a broad measure of SES that aims to capture economic and social metrics of wellbeing [25,26,27]. It is derived using a combination of material (e.g., education, income), social (e.g., dependency, partner status), and cultural (e.g., ethnicity, language) characteristics [28]. Additionally, social-interactive neighborhood elements such as disorder have emerged as important factors affecting child development [29]. Neighborhood disorder describes the climate of peace, safety, and law observance in a community, existing on a continuum that is indicated by visible cues of crime, social disorder (e.g., loitering, noise), and physical disorder (e.g., litter, property damage) [30]. Neighborhood literature has consistently demonstrated that place matters to development [24], even spurring research into the ways neighborhood characteristics affect learning and literacy [31]. Notably, studies have demonstrated that distal neighborhood environmental influences can impact child language skills in a pattern similar to more proximal individual characteristics [32] and sometimes to an even larger degree [33].

As evidence grows supporting the fact that neighborhoods are essential to holistic accounts of child development, there remains a limited understanding about the mechanisms underlying neighborhood effects on language. Due to its theoretical link with both environmental factors and individual mental processes, sleep is a candidate mechanism through which neighborhoods may operate on language. Regarding neighborhood factors and sleep, recent systematic reviews and meta-analyses have revealed that children living in more deprived and disorderly neighborhoods experience definitively worse sleep health than their peers [34,35]. In particular, neighborhood-level deprivation is associated with less total sleep hours and worse sleep quality [36,37,38], and higher neighborhood disorder is negatively associated with sleep duration and continuity [39,40]. Concerningly, neighborhoods of longstanding disadvantage predict even worse child sleep outcomes than neighborhoods with an equal but more recent history of low SES [41], which points to an urgent need to address inequities in developmental environments.

In terms of language and sleep, there is a large body of experimental literature demonstrating the benefits of sleep on language learning, particularly through facilitating memory processes [42,43]. Sleep supports diverse aspects of language skills including word production and recognition, grammar, and rule abstraction [42,44,45]. As such, researchers have highlighted the potential for targeting sleep to improve second language learning in adults [43]. Experimental studies testing the effects of sleep deprivation on child language learning are less common, though child sleep and memory processes, which support language, are relatively well-explored and understood [46]. Combined with research showing that sleep heightens children’s sensitivity to novel linguistic structures and is associated with increased word retention and vocabulary knowledge [47,48,49], sleep is a promising mechanism to explore in the relationship between neighborhoods and child language. Examining individual-level factors and their possible links with wider contextual influences aligns with recent efforts for a more comprehensive understanding of language development using advanced methodology [50].

Contemporary recommendations call for investigation into how, when, and for whom neighborhoods matter in terms of developmental outcomes [24,29,51]. Accordingly, our study aims to integrate existing theoretical and empirical links to further elucidate how neighborhood factors affect child language, and test sleep as a potential underlying mechanism through which distal environmental factors act on individual child outcomes. Using secondary data from the All Our Families cohort study, the primary objective of the current study is to examine the extent that neighborhood deprivation and disorder during pregnancy are associated with language skills when children are 5 years old. The influence of neighborhood factors during pregnancy has yet to be explored in relation to child language outcomes despite other perinatal influences being significantly linked to child development [52]. Aligned with preliminary evidence suggesting that developmental outcomes are similarly affected by family- and neighborhood-level influences [53,54], we hypothesized that neighborhood factors would be associated with child language development such that higher levels of deprivation and disorder would predict lower language scores. The secondary goal of the current study is to explore whether sleep duration during the first year of infancy represents a potential pathway through which neighborhood influences language development. Greater neighborhood deprivation and disorder were expected to be associated with fewer hours of consolidated sleep during the night and, in turn, poorer child language skills.

## 2. Materials and Methods

### 2.1. Participants and Procedures

The current investigation uses secondary data from the larger All Our Families (AOF) cohort study [55,56]. Ethics approval was obtained from the Conjoint Health Research Ethics Board (CHREB) at the University of Calgary for the AOF cohort study (REB13-0868) and the current analysis (REB16-1047). A total of 3388 pregnant women were recruited from May 2008 to December 2010 through primary healthcare offices, laboratory services, and community posters [56]. Eligibility criteria included being pregnant less than 25 weeks gestation, at least 18 years of age, able to complete questionnaires in English, and receiving prenatal care near Calgary, Canada. All participants provided informed consent prior to enrolment. Participants completed a battery of questionnaires before 25 weeks gestation (Q1); 34–36 weeks gestation (Q2); at 4 months (Q3) and 12 months (Q4) postpartum; and when children were 2 years (Q5), 3 years (Q6), and 5 years (Q7) of age.

### 2.2. Measures

#### 2.2.1. Participant Characteristics

Participants were asked to report sociodemographic data including age, marital status, education, household income, and ethnicity during the baseline (Q1) questionnaire. Participants also provided information about their psychosocial and physical health, delivery and birth outcomes, family history of language delays, child’s exposure to other languages, and neighborhood stability (i.e., moving).

#### 2.2.2. Neighborhood Deprivation

The Vancouver Area Neighborhood Deprivation Index (VANDIX) was used to measure neighborhood-level socioeconomic status during pregnancy, when postal code data were collected. The VANDIX is a census-based tool developed as a comprehensive examination of material (e.g., income, education, employment) and social (e.g., single-parent households) components of neighborhood deprivation pertinent to health outcomes [25,57]. Participant postal codes (Q1) were converted to latitudinal and longitudinal coordinates, which were mapped onto the City of Calgary community districts using the spatial join tool in ArcGIS Desktop (Version 10.6.1). Socioeconomic information for each identified neighborhood was obtained from the 2011 National Household Survey [58], accessed through the Calgary Community Data Consortium. In accordance with VANDIX protocol, seven socioeconomic factors were weighted, standardized, and summed to create VANDIX scores for each neighborhood, where higher scores indicate greater deprivation [25,57].

#### 2.2.3. Neighborhood Disorder

Calgary Police Services 2011 Community Crime Reports, which followed the Statistics Canada Uniform Crime Reporting guidelines [59], were accessed through the University of Calgary archives to create a neighborhood disorder index. Statistics for three indicators of disorder were available for each community district: objective social disorder (e.g., noise, threats, and general disturbance), physical disorder (e.g., fire, property damage), and crime (e.g., theft, break-and-enter, nondomestic assault, violence). An aggregate neighborhood disorder score was created by summing the number of reports for each indicator.

#### 2.2.4. Sleep

Child sleep duration was measured at 4 months (Q3) and 12 months (Q4) postpartum using a maternal report on a single-item question (“How many hours in a row does your baby usually sleep at night?”). Sleep duration is considered one of the most important dimensions and valid measurements of sleep health [60].

#### 2.2.5. Vocabulary

Child vocabulary at 24 months (Q5) was measured using the McArthur–Bates Communicative Development Inventories: Words and Sentences (CDI–WS) [61]. The CDI–WS is a parent-report instrument designed to measure language development for children from 16- to 30-months-old. It evaluates productive vocabulary through an inventory of 680 words spanning 22 semantic categories (e.g., animals, food and drink, body parts) and early grammar (e.g., the use of sentences). The CDI–WS has good overall diagnostic accuracy [62], shows high concurrent validity with scales measuring similar language proficiencies [63,64], and has high validity for parent report [65].

#### 2.2.6. Language

Language ability at 5 years (Q7) was measured using the General Communication Composite (GCC) score of the Children’s Communication Checklist–2 (CCC–2) [66]. The inventory was developed to assess child language skill as well as aspects of social language use. The CCC–2 contains 70 items across 10 scales that cover several language components including speech, syntax, coherence, and social relations [67]. The GCC is derived from summing scores of all 10 scales, where a low score suggests weak skills across all dimensions (i.e., lower overall ability). The CCC–2 has good internal consistency and test–retest reliability [67,68].

### 2.3. Data Analysis

Descriptive statistics and correlations for the main study variables were performed using SPSS 26. Multilevel modeling (MLM), with Mplus 8.1, was used to test neighborhood effects on child language skills, as it accounts for potential nonindependence of data from individuals in the same neighborhood [24]. Two-level random models were estimated where within-neighborhood variables were included at level 1 and between-neighborhood variables were included at level 2. Indirect effects of sleep duration were estimated using path analyses, where significance is indicted by a 95% confidence interval that does not cross zero [69]. Potential control variables considered included maternal age, ethnicity (coded as European Canadian = 1 or not = 0), education (coded as postsecondary = 1 or below = 0), household income (coded as >80 K = 1 or below = 0), family history of language delay (coded as mother, father, and/or sibling = 1 or not = 0), infant sex (coded as male = 1 or female = 0), preterm birth (coded as <37 weeks = 1 or more = 0), small for gestational age (coded as <10th percentile = 1 or more = 0), neighborhood stability (coded as any moves between birth and 3 years old = 1 or none = 0), vocabulary at 2 years old, and regular exposure to other languages at 5 years (coded as yes = 1 or 0 = no) based on previous associations with language development [11,12,70,71,72,73,74]. Potential control variables that were significantly correlated with language skills at 5 years were included as level 1 covariates in MLM and path analyses. Full information maximum likelihood (FIML) estimation was used to handle missing data [75].

## 3. Results

### 3.1. Sample Description

Of the 3388 total participants enrolled in the study, 1 withdrew their data and 36 were excluded for twin births. An additional 907 participants could not be geocoded for the neighborhood analyses because they did not provide postal codes or lived outside of Calgary city limits during the first pregnancy questionnaire (Q1). The final analyses included 2444 women across 192 neighborhoods, with an average of 12.73 participants per neighborhood.

At the time of enrollment (Q1), the average age of participants was 30.78 (SD = 4.49) and the majority were European Canadian (13.1% Asian Canadian, 2.3% Latinx Canadian, 1.5% African Canadian, 1.5% Arab Canadian, 0.8% Indigenous, 3.6% Mixed/Other), reported a yearly household income of over $80,000 CAD, had completed postsecondary education, and reported being partnered. Characteristics of the final sample are presented in Table 1.

### 3.2. Bivariate Correlations

Correlations between neighborhood variables, sleep duration, and language are reported in Table 2. Neighborhood deprivation and disorder were significantly correlated. Both deprivation and disorder were significantly associated with shorter child sleep duration at 1 year postpartum (Q4), as well as lower General Communication Composite scores at age 5 (Q7). Child sleep duration at 1 year postpartum was also significantly positively associated with General Communication Composite scores and was therefore retained for MLM and path analyses. 

Correlations between predictors and potential covariates revealed that the General Communication Composite score was significantly associated with maternal race/ethnicity (*r* = 0.114, *p* < 0.001), education (*r* = 0.121, *p* < 0.001), household income (*r* = 0.110, *p* < 0.001), family history of language delay (*r* = −0.122, *p* < 0.001), infant sex (*r* = −0.141, *p* < 0.001), preterm birth (*r* = −0.074, *p* < 0.001), and vocabulary at 2 years (*r* = 0.343, *p* < 0.001). These variables were thus included as level 1 covariates in the subsequent MLM and path analyses. All other potential covariates including age, neighborhood stability, and regular exposure to other languages were not statistically significant and excluded from further analyses. 

### 3.3. Multilevel Modeling

The intraclass correlation (ICC) indicated that 1.7% of the total variation in General Communication Composite language scores was attributable to between-neighborhood differences. Multilevel model results for within- and between-neighborhood variables are presented in Table 3, where parameter estimates represent change in the General Communication Composite. After controlling for level 1 individual and family characteristics, both neighborhood deprivation and disorder during pregnancy emerged as significant negative predictors of General Communication Composite scores. Specifically, for the neighborhood deprivation scores, every 1 standard deviation (approximately 2.9 units) increase in the VANDIX was associated with a 0.567 decrease in mean child General Communication Composite scores; for the unstandardized disorder scores, every additional police report (crime, social or physical disorder) in a neighborhood was associated with a 0.001 decrease in mean language scores (i.e., every 1 standard deviation increase in police reports associated with a 0.684 decrease in mean child General Communication Composite scores).

In terms of individual- and family-level factors, having a mother who identified as European Canadian or had completed postsecondary education was associated with higher child General Communication Composite scores whereas a family history of language delay was associated with lower scores. Infant sex was related to language, with girls having higher General Communication Composite scores than boys. A larger vocabulary at 2 years was also associated with higher General Communication Composite scores at 5 years. None of the other level 1 covariates significantly predicted child General Communication Composite language scores within neighborhoods. The multilevel model indicated that the relationship between child sleep duration at one year postpartum (Q4) and language scores at 5 years across neighborhoods did not reach statistical significance. However, indirect effects can be present and interpreted regardless of whether the constituent paths are statistically significant [76].

### 3.4. Indirect Effects

Separate path analyses were conducted using MLM in order to test potential indirect effects of neighborhood deprivation and disorder through sleep duration at Q4. After adjusting for covariates, a significant indirect effect of neighborhood disorder on language development was found at 5 years through sleep duration at 1 year (*ab* = −0.001, *p* = 0.029, 95% *CI*: −0.002, 0.000). In other words, increased neighborhood disorder was associated with shorter sleep duration, which in turn was related to lower child General Communication Composite scores. No significant indirect effects were found for neighborhood deprivation through infant sleep duration at Q4 (*ab* = −0.150, *p* = 0.117, *CI*: −0.337, 0.037).

## 4. Discussion

### 4.1. Summary and Interpretation of Findings

The present study investigated the influence of neighborhood factors during pregnancy on child language outcomes at 5 years of age using secondary analysis of the All Our Families cohort study. Multilevel modeling indicated that neighborhood deprivation and disorder were associated with poorer language skills at age 5. Path analyses suggested a small indirect effect of neighborhood disorder, but not neighborhood deprivation, on language at 5 years through shorter sleep duration at 12 months. These results highlight the subtle interplay of early environmental factors on child development.

As expected, higher levels of neighborhood deprivation and disorder during pregnancy were associated with lower language scores among children at 5 years of age. Our results build on evidence that neighborhood-level factors can exert effects on child development when controlling for family-level covariates [77,78]. Indeed, we found that neighborhoods have a modest effect on child language development beyond known predictors of child language including family history of language delay, infant sex, and early vocabulary [11,73]. Neighborhood factors accounted for approximately 2% of variance in child language scores in the present study, which is aligned with another recent Canadian study observing that neighborhoods were associated with 3% of the variability in children’s language competency [79]. Comparably small neighborhood effects on developmental outcomes have also been observed in U.S. and British populations [80,81]. However, modest environmental effects are integral to comprehensive accounts of development, particularly when examining population-level health outcomes of a diverse populace [57,82]. This is especially relevant in the context of recent claims of ethnocentrism regarding previously accepted predictors of child language skill and language learning models (e.g., the ‘language gap’ of word exposure between higher- and lower-SES families [15,83,84]). The present study highlights that further investigation is required to determine accurate and precise mechanisms through which SES influences developmental outcomes. 

The current results are the first to suggest that neighborhood during pregnancy may be implicated in the development of language abilities. Deprivation and disorder are commonly cited as predictors of later childhood influences on development [85,86], but studies have not specifically focused on perinatal environmental factors. Our analyses revealed that neighborhood stability between the perinatal period (i.e., when postal code data was collected) and three years postpartum was not significantly correlated with child language scores. Most women who move within the first year postpartum tend to stay in the same census tract areas (i.e., move to neighborhoods with similar levels of poverty [87]), which may explain why language ability appears unaffected by early life relocation in the context of neighborhood influences. However, given that almost one quarter of mothers in our study moved during their first three years postpartum, the lack of association may offer preliminary evidence that fetal development may be sensitive to neighborhood-level factors. Adverse developmental environments have previously been linked with neurobiological changes that predict worse health outcomes in adulthood including affective, antisocial, and other mental disorders [88,89]. The current findings highlight the potential importance of assessing environmental vulnerabilities as early as pregnancy. 

Aligned with previous reports [34], the current study found a significant negative correlation between neighborhood deprivation and infant sleep duration; however, the indirect effect on language development through sleep duration was not significant. Past studies may have confounded neighborhood deprivation and disorder as they did not measure them separately (e.g., Grimes et al., 2019; [37]). Our results did suggest sleep duration as a possible underlying mechanism of neighborhood disorder, such that more reports of disorder during pregnancy were associated with shorter sleep duration at 12 months and, in turn, poorer child language skills at 5 years. This finding is in line with recent reviews suggesting that neighborhood disorder affects child sleep in a social-interactive nature [29]—that is, neighborhoods can influence children through parental reactions to their environment. For example, mothers who perceive their neighborhood to be high in disorder are less likely to create responsive (e.g., communicative, affective) and organized (e.g., in physical space or routine) home environments [90], which can affect child sleep. Since sleep is observed to support vocabulary knowledge, improvements in grammatical structure, and maintenance of new linguistic rules [44,45,47], it follows that poorer sleep would negatively affect language development. 

Taken together, our findings support the emerging consensus that neighborhood factors persistently exert small but significant effects on child developmental outcomes across a range of locations and public policies [91]. A dynamic approach to supporting optimal development should integrate attention to both wider contextual factors, such as infusing resources into deprived and disordered neighborhoods, and more proximal influences, such as family behaviors. The current finding that sleep represents a potential path through which neighborhood factors influence language points to a target for supporting children’s development. Child development is highly sensitive to environmental factors during the first five years of life [92], meaning that this period of development provides a prime opportunity to effectively mitigate potentially detrimental influences. Early interventions are efficacious for equalizing the language skill of children from differing SES backgrounds, and the impacts are long-lasting [93,94]. Our study results offer child sleep as a feasible and accessible method to ameliorate potential negative effects of neighborhood influences, while reinforcing calls to develop large-scale solutions that are necessarily more time- and resource-intensive. 

### 4.2. Strengths and Limitations

The large sample size and prospective design of the All Our Families cohort study enabled a robust test of the unique associations of neighborhood factors on child language beyond several established individual- and family-level factors. Specifically, the detailed family history collected from participants allowed an analysis that could disentangle family effects on child language skill from the developmental environment. Beyond family-level factors, longitudinal studies are particularly effective for discerning neighborhood influences because they are sensitive to self-selection in neighborhood membership [54]; further, use of multilevel modeling accounted for internal heterogeneity of neighborhoods while testing for nested factors of development [18,24]. Additionally, the measure of neighborhood deprivation was derived from census tract information, which is more objective and stable over time than individual measures in [19]. We acknowledge that while our measures and analyses were carefully designed, future experimental studies would enhance our understanding of precisely how neighborhoods affect child development. To date, only observational studies have been used to explore this issue [29]. Natural experimental designs (i.e., manipulating a feature of interest in an existing neighborhood) would be an innovative way to test an intervention while clarifying the relationship between broader environment and developmental outcomes.

Findings from the current investigation should also be interpreted with some limitations in mind. The results are from families living within a system of universal health care, routine screening, and public education, which is reflected in the limited range of neighborhood deprivation. We were also constrained by the availability of sleep variables in the cohort study. Further investigation may benefit from objective measures of sleep (e.g., actigraphy), as subjective measures have shown less-pronounced effects in comparison [34,35]. Assessing components of sleep health beyond duration (e.g., efficiency, timing, satisfaction; [60]) would also further elucidate the mechanistic nature of sleep. There is evidence that child sleep trajectories and sleep-dependent learning pathways can change in early childhood [47,95], so multiple timepoints could be helpful for extricating more-precise intervention targets. Similarly, since 23% of our study participants moved in their first three years postpartum, measuring neighborhood deprivation and disorder after pregnancy may more clearly distinguish which early environmental factors are important for language development. While most women who move within the first year postpartum tend to stay in the same census tract areas (i.e., move to neighborhoods with similar levels of poverty [87]), additional measures of neighborhood factors are necessary to ensure that perinatal environment is representative of early infancy environment. Finally, supplementary language outcome measures would corroborate the current findings. The present study used the well-validated CCC-2 parent report of language, which is robust compared with equivalent tests of pragmatic skill [66,96,97]. Since social language deficits can be detrimental for a child’s school and leisure functioning [98], comprehensive assessment of language skills could offer an extension and/or clarification about specific language components that are affected by developmental contexts.

## 5. Conclusions

The current study corroborates expanding literature that demonstrates the essential role of neighborhoods in holistic accounts of child development. It is among the first to examine child language in the context of perinatal neighborhood environment, providing preliminary evidence for the importance of assessing environmental vulnerabilities as early as pregnancy. Our findings suggest that child outcomes may be affected by distal neighborhood-level factors beyond individual and family characteristics. These findings highlight the need to address developmental environment disparities at social and interpersonal levels in addition to the traditional focus on family influences. The current study also heeded recommendations for research to investigate how specifically neighborhoods affect child development. We proposed a novel link and found evidence suggesting that sleep is a potential mechanism through which neighborhoods impact child outcomes. Large-scale changes such as resource equalization and community development require considerable time and resources. This study offers child sleep as an accessible target for interventions to buffer the effects of adverse developmental environments.

## Figures and Tables

**Table 1 brainsci-12-00223-t001:** Sample characteristics.

	*n* (%)	M (SD)
Family		
Maternal age (years)		30.78 (4.49)
Maternal race/ethnicity (European Canadian)	1878 (76.8)	
Maternal education (post-secondary)	1848 (75.6)	
Household income (>80 K)	1652 (67.6)	
Family history of language delay	102 (4.2)	
Neighborhood stability (moved birth to 3 years)	572 (23.4)	
Individual		
Infant sex (male)	1206 (49.3)	
Preterm birth (<37 weeks)	164 (6.7)	
Small for gestational age (<10th percentile)	231 (9.5)	
Vocabulary at 2 years		291.76 (283.00)
Regular exposure to other language(s)	563 (23.0)	

**Table 2 brainsci-12-00223-t002:** Descriptive statistics and correlations for main variables.

		1	2	3	4	5
Neighborhood (Level 1)						
1	Deprivation	-				
2	Disorder	0.197 **	-			
Sleep (Level 2)						
3	Duration at Q3	0.005	−0.029	-		
4	Duration at Q4	−0.092 **	−0.079 **	0.278 **	-	
Language (Level 2)						
5	GCC	−0.144 **	−0.083 **	0.028	0.073 *	-
Mean		−6.67	591.67	6.36	9.23	87.4
SD		2.86	684.44	2.56	2.83	14.78

Note: Level 2 = between neighborhoods, Level 1 = within neighborhoods (individual), GCC = General Communication Composite. * *p* < 0.05, ** *p* < 0.01.

**Table 3 brainsci-12-00223-t003:** Multilevel model results for neighborhood predictors of language at 5 years.

	Estimate	SE	*p*	95% CI
Intercept		6.038	<0.001	80.738, 104.405
Covariates (Level 1)				
Maternal race/ethnicity (European Canadian)	2.912	1.018	0.004	0.916, 4.908
Maternal education (postsecondary)	3.082	0.857	<0.001	1.403, 4.761
Household income (>80 K)	1.406	0.828	0.090	−0.218, 3.029
Family history of language delay	−5.492	1.673	0.001	−8.771, −2.213
Infant sex (male)	−3.244	0.712	<0.001	−4.639, −1.849
Preterm birth (<37 weeks)	−1.632	1.378	0.236	−4.333, 1.070
Vocabulary at 2 years ^a^	0.024	0.003	<0.001	0.019, 0.029
Sleep (Level 2)				
Sleep duration (Q4)	1.259	0.654	0.054	−0.023, 2.542
Neighborhood (Level 2)				
Deprivation	−0.567	0.132	<0.001	−0.826, −0.307
Disorder	−0.001	0.000	0.001	−0.001, 0.000

^a^ Mean centered.

## Data Availability

The data presented in this study are available through the Secondary Analysis to Generate Evidence (SAGE) repository platform administered by PolicyWise for Children and Families (https://allourfamiliesstudy.com/sage//; last accessed 28 November 2021).

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
