# Peer review of "Does Where You Live Predict What You Say? Associations between Neighborhood Factors, Child Sleep, and Language Development"

_brainsci, 2022, doi:10.3390/brainsci12020223_

Round 1
Reviewer 1 Report
Thank you for the opportunity to read and review this manuscript.
The manuscript is relatively well written, and it may be of interest to the readership of the Brain Sciences journal. However, there are several issues that may need to be addressed.
In the introduction section, a more detailed review of the literature related to the relationships among neighbourhood factors, child sleep, and child language development is needed to highlight the necessity/rationale for this current study.
There was a nice discussion of the findings in Sections 4.1 and 4.2. However, the authors need to reiterate the significance of the study in the conclusion section.
Author Response
Comment: The manuscript is relatively well written, and it may be of interest to the readership of the Brain Sciences journal. However, there are several issues that may need to be addressed. In the introduction section, a more detailed review of the literature related to the relationships among neighbourhood factors, child sleep, and child language development is needed to highlight the necessity/rationale for this current study.
Response: Thank you for this suggestion on how to strengthen the rationale for our manuscript. We have completed a more detailed literature review and expanded the background section for our manuscript, including several relevant reviews (e.g., Kopasz 2010, Pace 2017, Schimke 2021, Schreiner 2017). Additionally, we have restructured parts of the introduction to highlight the theoretical and empirical links we are drawing from to rationalize our research questions. One of our primary goals was to use a bioecological model of development as a framework for our study, as language learning is complex and inherently linked with broader structures of influence such as culture. Your comment has helped us clarify how our variables draw from and fit into the model.
Comment: There was a nice discussion of the findings in Sections 4.1 and 4.2. However, the authors need to reiterate the significance of the study in the conclusion section.
Response: Thank you for this suggestion. We have reiterated the significance of the study in the conclusion section.
Reviewer 2 Report
- This is a meaningful study.
- The paper is well written.
- The introduction is helpful to understand the goal of the research.
- In the conclusion, it is suggested that adding clearly stating the application and the contribution to the field.
Author Response
Comment: This is a meaningful study. The paper is well written. The introduction is helpful to understand the goal of the research. In the conclusion, it is suggested that adding clearly stating the application and the contribution to the field.
Response: Thank you for your constructive comments. We have reworked our conclusion to clearly state the contribution to the field and included possible applications.
Reviewer 3 Report
I greatly enjoyed reading this paper, which is both intellectually stimulating and very carefully written. I like this paper, because it manages to combine convincingly the theory part with the data analysis. The balance between the two parts is just right. I suggest the authors to add a caveat on the need for future research to explore these issues.
Author Response
Comment: I greatly enjoyed reading this paper, which is both intellectually stimulating and very carefully written. I like this paper, because it manages to combine convincingly the theory part with the data analysis. The balance between the two parts is just right. I suggest the authors to add a caveat on the need for future research to explore these issues.
Response: Thank you for your thoughtful comments. We have added a caveat in the ‘strengths and limitations’ section that future experimental (vs. observational) research can further support the ideas we developed in the current paper.
Round 2
Reviewer 1 Report
Thank you for your revisions. Your manuscript has been much improved, and I believe that it is ready for publication.